# Lossy compression of earth system model data based on hierarchical tensor with Adaptive-HGFDR (V1.0)

Zhaoyuan Yu[1,2], Dongshuang Li[3,4], Zhengfang Zhang[1], Wen Luo[1,2], Yuan Liu[1], Zengjie Wang[1], Linwang Yuan[1,2,*]

[1]Key Laboratory of Virtual Geographic Environment, Ministry of Education, Nanjing Normal University, Nanjing, China,
[2]Jiangsu Center for Collaborative Innovation in Geographical Information Resource Development and Application, Nanjing, China,
[3]Jiangsu Key Laboratory of Crop Genetics and Physiology/Jiangsu Key Laboratory of Crop Cultivation and Physiology, Agricultural College of Yangzhou University, Yangzhou, China,
[4]Jiangsu Co Innovation Center for Modern Production Technology of Grain Crops, Yangzhou University, Yangzhou, China

*Correspondence to*: Linwang Yuan (email: yuanlinwang@njnu.edu.cn)

**Abstract.** Lossy compression has been applied to the data compression of the large-scale earth system model data (ESMD) due to its advantages of a high compression ratio. However, few lossy compression methods consider both the global and local multidimensional coupling correlations, which could lead to the information loss in data approximation of lossy compression. Here, an adaptive lossy compression method, Adaptive-HGFDR is developed on the foundation of a stream compression method for geospatial data, Blocked Hierarchical Geospatial Field Data Representation (Blocked-HGFDR). Yet, the original Blocked-HGFDR method is improved from the following perspectives. Firstly, the original data are divided into a series of data blocks with more balanced size to reduce the effect of the dimensional unbalance of ESMD. Then based on the mathematical relationship between the compression parameter and compression error in Blocked-HGFDR, the control mechanism is developed to determine the optimal compression parameter for the given compression error. By assigning each data block independent compression parameter, Adaptive-HGFDR can capture the local variation of multidimensional coupling correlations to improve the approximation accuracy. Experiments are carried out based on the Community Earth System Model (CESM) data. The results show that our method has higher compression ratio and more uniform error distributions, compared with ZFP and Blocked-HGFDR. For the compression results among 22 climate variables, Adaptive-HGFDR can achieve good compression performances for most flux variables with significant spatio-temporal heterogeneity and fast changing. This study provides a new potential method for the lossy compression of the large-scale earth system model data.

## 1 Introduction

Earth System Model Data (ESMD), which comprehensively characterize the spatio-temporal changes of earth system with multiple variables, are presented as multidimensional arrays of floating-point numbers (Kuhn et al., 2016;Simmons, 2016). With the rapid development of earth system models in finer computational grids and growing ensembles of multi-scenario

simulation experiments, ESMD have shown an exponential increase in data volume (Nielsen et al., 2017; Sudmanns et al.,
2018). The huge data volume brings considerable challenges to the data computation, storage, and analysis on ordinary PCs,
which will further limit the research and application of ESMD. Lossy compression, which focuses on saving large amounts
of data space by approximating the original data, is considered as an alternative solution to meet the challenge of the large
data volume(Baker et al., 2016; Nathanael et al., 2013). However, ESMD, as a comprehensive interaction of earth system
variables at different aspects of space, time, and attributes, show the significant multidimensional coupling
correlations(Runge et al., 2019; Mashhoodi et al., 2019; Shi et al., 2019). The mixture of different coupling correlations then
leads to complex structures, such as the uneven distribution, spatially nonhomogeneity and temporally nonstationary, which
increases the difficulties in accurately approximating data in lossy compression. Thus, developing a lossy compression
method that could adequately explore the multidimensional coupling correlations is an important way to reduce the
compression error(Moon et al., 2017).
Predictive and transform methods are two of the most widely used lossy compression approaches in terms of how the data is
approximated. Predictive lossy compression predicts the data with parametric functions, and the compression is achieved by
typically retaining (and encoding) the residual between the predicted and actual data value. For example, NUMARCK learns
emerging distributions of element-wise change ratios and encodes them into an index table to be concisely
represented(Zheng et al., 2016). ISABELA applies a preconditioner to seemingly random and noisy data along spatial
resolution to achieve an accurate fitting model for the data compression(Lakshminarasimhan et al., 2013). In these methods,
the multidimensional ESMD are processed as low dimensional sequences or series without considering the multidimensional
coupling correlations. SZ, one of the most advanced lossy compression methods, features adaptive error-controlled
quantization and variable-length encoding to achieve the optimized compression (Ziv and Lempel, 2003). In SZ, a set of
adjacent quantization bins are used to convert each original floating point data value to an integer along the first dimension
of the data based on its prediction error (Di et al., 2019). With a well-designed error control mechanism, SZ can achieve the
uniform compression error distribution. However, SZ predicts the data point only along the first dimension, and it is not
designed to be used along the other dimensions or use a dynamic selection mechanism for the dimension (Tao et al., 2017).
This makes the data inconsistency problem of SZ, where the same ESMD with different organization orders can capture
different multidimensional coupling correlations, and further produce different compressed data.
Transform methods, reduce data volumes by transforming the original data to another space where the majority of the
generated data are small, such that the data compression can be achieved by storing a subset of the transform coefficients
with a certain loss in terms of the user's required error (Diffenderfer et al., 2019; Andrew et al, 2020). One example is the
image-based method, which slices ESMD from different dimensions into separate images, and each image is then
compressed by feature filtering with wavelet transformation or Discrete Fourier Transform (Taubman and Marcellin, 2002).
As the compression is applied to the single image slice, the coupling correlations among multiple dimensions are not always
well utilized. More advanced method like ZFP splits the original data into small blocks with an edge size of 4 along each
dimension, and compresses each block independently via a floating-point representation with a single common exponent per

block, an orthogonal block transform, and embedded encoding(Tao et al., 2018). In ZFP, the multidimensional coupling correlations are integrated by treating all dimensions as a whole through multidimensional blocking. In each block, ZFP converts the high dimensional data into matrics, which yet flattens the data and partially destroys the internal correlations among multiple dimensions. Additionally, with only a single common exponent used in each block, it is inadequate to capture the local variation of the correlations. Thus, the ZFP method is extremely effective in terms of data reduction and accuracy for smooth variables, but are unsurprisingly challenged by variables with abrupt value changes and ranges spanning many orders of magnitude, both of which are common in ESMD outputs (Baker et al., 2014).

Most of the current existing lossy compression methods, including predictive and transform lossy compression methods, integrate the multidimensional coupling correlations to the process of data approximation on the foundation of mapping multidimensional data into low dimensional vector or matrics(Wang et al., 2005). Few of these methods directly process multidimensional ESMD as a whole. For instance, current predictive methods usually split the original data into a series of local low-dimensional data, then predict each local data respectively. In this way, the splitted data obtained by different split strategies could capture the different coupling correlations, which further lead to the inconsistent compressed results for the same data. Transform methods map the original data to the small space by removing the redundant coupling correlations. Most of these methods have already considered the coupling correlations in the global region. However, each local region still utilizes the data splitting that destroys the local coupling correlations, which result in the weak compression performance for the ESMD with strong local variations. Therefore, constructing the lossy compression method that integrates both global and local coupling correlations from the perspective of multiple dimensions, is helpful to improve the performance of lossy compression for ESMD.

Recently, the tensor-based decomposition methods, such as the Canonical Polyadic (CP) , Tucker and hierarchical tensor decomposition, have been introduced to the compression of the multidimensional data(Bengua et al., 2016; Jing et al., 2014). The tensor decomposition, which exploits the data features along with each mode and the corresponding coupling relationship by considering the multidimensional data as a whole, can estimate the intrinsic structure of ESMD ignored in the metric model. The core motivation behind the tensor-based decomposition is to eliminate the inconsistent, uncertain, and noisy data without destroying the intrinsic multidimensional coupling correlation structures (Kuang et al., 2018; Du et al., 2017). Among these methods, the hierarchical tensor decomposition could achieve higher quality at large compression ratio than traditional tensor methods through extracting data features level by level (Wu et al., 2008). Yuan et al (2015) designed an improved hierarchical tensor method (Blocked-HGFDR) to compress geospatial data with a hierarchical tree structure, showing the obvious advantages in the compression accuracy and compression efficiency. This hierarchical-tensor based method utilizes the multidimensional coupling correlations to approximate the original data by treating all dimensions as a whole, which can largely reduce the information loss in lossy compression. In Blocked-HGFDR, each local data own the same compression parameter and the global average error is used to control the capture of the global multidimensional coupling correlation. Since ESMD are always spatio-temporal heterogeneous where the coupling correlations are various in each local region, the same compression parameter applied to each local data results in the insufficient capture of the local

coupling correlation. Although the global average error is relatively small, the obtained results tend to a certain "average"
within the each local data, which may make the local compression error very large so as to bring the bias to the data
approximation.
In this paper, the lossy compression for ESMD is developed based on the Blocked-HGFDR. We firstly construct a division
strategy that divides the original data into a series of data blocks with relatively balanced dimension. Then the parameter
control mechanism is designed to assign each data block the independent compression parameter under the given
compression constraint. After that, Blocked-HGFDR is applied to each data block to achieve the lossy compression.
Experiments on climate simulation dataset with 22 variables are carried out to evaluate the performance and applicability of
the methods in ESMD compression. The remainder of this paper is organized as follows. Section 2 introduces the basic ideas
about developing Adaptive-HGFDR. Section 3 discusses the block mechanism, the relationship between the compression
parameter and compression error, and the fast search algorithm. Section 4 uses the temperature data to verify that the method
can obtain adaptive rank under the accuracy constraint. Section 5 discusses the effectiveness and computational efficiency of
the method, as well as the results.

## 113   2 Basic idea

The lossy compression of ESMD should comprehensively consider the characteristics of ESMD. Firstly, since ESMD
have multiple variables, the compression parameter of an ideal lossy compression should be simple and can be flexibly
adjusted according to the corresponding variables of ESMD. Secondly, since the acceptable error of different variables
in ESMD is different, for example, the error of wind speed is very different from that of temperature. So an ideal lossy
compression should be able to select adaptively compression parameters for the acceptable error range of different
variables. Considering that Blocked-HGFDR has simple compression parameter, it can be used for the lossy
compression of ESMD. Thirdly, since many variables of ESMD have spatio-temporal heterogeneity, the corresponding
coupling correlations are variate within the local region. Thus, the correlations in both global and local region should
be well integrated in lossy compression to improve the approximation accuracy.
In order to adequately integrate the multidimensional coupling correlations and adaptively select the compression
parameter in Blocked-HGFDR, there are two issues to be considered. The first issue is the dimensional unbalance of
ESMD. For instance, the data accumulated in the temporal dimension is typically longer than that in the spatial
dimension for a spatio-temporal series with long observations. Since the tensor decomposition method treats each
dimension equally that ignores the dimensional unbalance, it is difficult to accurately approximate data with
unbalanced dimensions. Thus, it is better to split the original data into small local data blocks with the more balanced
dimension structure, and then applying the tensor decomposition to each local data individually can reduce the
approximation bias caused by the dimensional unbalance. The second issue is the parameters selection under the given
compression constrains. Since the coupling correlations of ESMD vary within local regions, for the given compression
constrains such as the maximum compression error, the compression parameter of different variables or data blocks
should be selected flexibly according to the corresponding data characteristic, so as to well capture the local variation
of the coupling correlation to improve the approximation accuracy. Therefore, based on the mathematical relationship
between the compression error and the compression parameter in Blocked-HGFDR, a control mechanism, which can
adjust the compression parameter according to the accuracy demands should be developed.
Based on the above considerations, our methods, Adaptive-HGFDR, is developed according to the following three
procedures (Figure 1). Procedure 1: Splitting the original ESMD into small data blocks. In this procedure, the
dimension to split the data and the optimal size of the data block is determined by conducting different combinations
of data blocking in terms of the dimension and block counts. Procedure 2: Conducting the relationship between
compression error and compression parameter. In order to obtain a uniform distribution of the compression error for
each data block, an empirical relationship between the compression error and the rank value is established, where the
rank value of each data block can be adjusted at any given compression error. Procedure 3: Adaptive searching for the
optimal compression parameter. A binary search method is used to search the optimal compression parameter, which is
updated with a parameter control mechanism until the compression error meets the given constraint.

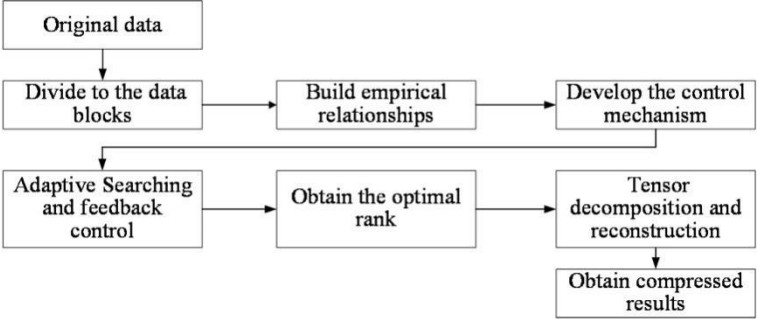



**Figure 1. Overall framework of the basic idea.**
**3 Method**
**3.1 Block hierarchical tensor compression**
EMSD is a multidimensional array. It can be seen as a tensor with the spatio-temporal references and the associated
attributes. Without loss of generality, a three-dimensional tensor can be defined as $Z \in \mathbb{R}^{I \times J \times K}$ (Suiker and Chang, 2000),
where $I$, $J$, and $K$ are values that represent the number of grids along the dimensions of longitude, latitude, and time (or
height), respectively. These dimensions are always unbalanced due to the different spatial and temporal resolutions. So, the
data block is introduced to reduce the impact of dimension unbalance on the data compression.


**Definition 1 Data block**
For the spatio-temporal data $Z \in \mathbb{R}^{I \times J \times K}$, it can be considered as composed of a series of local data with the same spatio-
temporal reference. Here, each local data is defined as the data block as follow:
$$part(Z, n) = \{C_1, C_2, \cdots, C_n\} \tag{1}$$
Here, $part(\ )$ is the function that divides the original tensor $Z$ into a series of data block $\{C_i\}_{i=1}^{m}$, each data block $C_i$ includes
local spatial and temporal information, and $n$ is the number of data blocks. Compared with the original data, the dimensions
of these data blocks are smaller and more balanced. For the divided data blocks, in order to adequately capture the
multidimensional coupling correlation, the key point is how to determine the compression parameter according to the given
compression error.

**Definition 2 Blocked-HGFDR**
Based on the divided data blocks, Yuan et al.(2015) proposed the Blocked-HGFDR method based on the hierarchical tensor
compression. In this method, the hierarchical tensor compression is applied to each block, then the hierarchical tensor
compression of each data block is obtained by selecting the prominent feature components and filtering out the residual
structure. This method utilizes the hierarchical structure of data features, greatly reducing data redundancy, and thereby
achieving the efficient compression of the amount of spatio-temporal data (Yuan et al., 2015). The overall compression of
Blocked-HGFDR can be formulated as:
$$\begin{cases} H(A) = (U_R \otimes U_{R-1} \otimes \cdots \otimes U_1) \tilde{B}_L \tilde{B}_{L-1} \cdots \tilde{B}_1 B_{12\cdots R} + \text{r}es \\ \quad \tilde{B}_j = B_{p_{L_j}} \otimes \cdots \otimes B_{p_L} \qquad j = \{1, 2, \ldots, L\} \end{cases} \tag{2}$$
Similar to the prominent components obtained by SVD for two-dimensional data(Yan et al., 2019), the matrix $U_R$ and the
sparse transfer tensor $B_R$ are considered to be the r-th component of a third-order tensor in each dimension, respectively,
where $R$ denotes the number of multi-domain features. The residual tensor, $\text{r}es$, in Eq. (2) denotes the information not
captured by the decomposition model, and $(U_R \otimes U_{R-1} \otimes \cdots \otimes U_1) \tilde{B}_L \tilde{B}_{L-1} \cdots \tilde{B}_1 B_{12\cdots R}$ in Eq. (2) is the reconstructed r-th core
tensor and feature matrix(Grasedyck, 2010; Song et al.,2013).
**3.2 Adaptive selection of parameter and solution**
Considering that the distribution characteristic of each divided data block is different (Hackbusch and Kühn, 2002), the key
to adequately capture the multidimensional coupling correlations in Blocked-HGFDR is to adaptively select the compression
parameter for each local data respectively according to the given compression error. So the key step is to construct
controlling mechanism based on the relationship between the compression error and compression parameter. Thus, the
following terms are defined.

**Definition 3 The controlling mechanism.**
In Blocked-HGFDR, the relationship between the compression error and compression parameter (*Rank*) is given as
$\varepsilon = a\, Rank^{-\beta}$ (Yuan et al., 2015), thus the controlling mechanism to determine the compression parameter of each block data
should be the rank value closest to the given compression error as follows:
$\quad \varepsilon = a\, Rank^{-\beta} \le \varepsilon_{Given}$ (3)
$\varepsilon_{Given}$ is the given compression error that depends on different application scenarios; $a, \beta$ are the coefficients depended on the
structure and complexity of the data, which can be obtained by the simulation experiment for actual data.
In Blocked-HGFDR, the relationship between the compression ratio ($\varphi$) and compression parameter (*Rank*) is given as
follows:
$\quad \varphi = \dfrac{datasize}{aRank^{3} + bRank^{2} + cRank + d}$ (4)
As shown in Eqs. (2), (3), and (4), in Blocked-HGFDR, with rank decreasing, the compression ratio of Blocked-HGFDR
increases, and the compression error also increases. In Blocked-HGFDR, the rank value of different blocks is fixed, which
results in the fluctuation of the compression error in the specific dimension. Since the structure of each block is different, the
compression parameter of each data block should be determined independently according to the given compression error.
Considering that the actual compression error may not strictly satisfy the given value, the optimal parameter is selected as
the minimum $Rank$ in which the obtained compression error is close to the given one.

To find the optimal parameter for data block $C_i$, with the above constructed controlling mechanism, the binary search
algorithm based on dichotomy is constructed. That means before adjusting the rank each time, the optimal rank
corresponding to the given compression error is constantly approached in half by reducing the selection interval by half of
the rank. The algorithm is implemented as follows:

---

**Algorithm:** the optimal parameter search algorithm based on dichotomy

---

**Input:** data block $C_i \in \mathbb{R}^{Q \times W \times E}$; given compression error $std\_err$;

**Output:** the optimal parameter $R\_Opt$

**Function Description:** $EvalErr(C_i, r)$ is used to calculate the error of hierarchical tensor SVD of $C_i$ at rank $r$ based

---

on Eqs. (4) and (6). *Round()* is the rounding function; *Max()* is the function which taking the maximum value

1: $R\_Max = Max(Q,W,E)$, $R\_Min = 0$

2: $R\_Mid = Round(\dfrac{R\_Max + R\_Min}{2})$

3: $err = EvalErr(C_i, R\_Mid)$

4: While ($err != std\_err$ && $R\_Max > R\_Min$)

5:     If ($err > std\_err$)

6:         $R\_Min = R\_Mid + 1$

7:     Else

8:         $R\_Max = R\_Mid - 1$

9:     End If

10:     $R\_Mid = Round(\dfrac{R\_Max + R\_Min}{2})$

11:     $err = EvalErr(C_i, R\_Mid)$

12: End While

13: Return ($R\_Opt = R\_Mid$)


During the whole algorithm, the function *EvalErr(C_i,r)* is the computing intensive function that could be the performance
bottleneck. If we consider a calculation of *EvalErr(C_i,r)* as one meta calculation, the complexity of the traditional traversal
method is $O(n)$. When introducing the dichotomy optimization, the complexity can be reduced to $O(\log n)$ (Cai et al., 2012).
**4 Case study**
**4.1 Data description and experimental configuration**
In this paper, data produced by Community Earth System Model are used as the experimental data to evaluate the
compression performance of Adaptive-HGFDR, which can be obtained from Open Science Data Cloud in NetCDF (Network
Common Data Form) format (http://doi.org/10.5281/zenodo.3997216). The data set includes air temperature data (T) stored
as a $1024 \times 512 \times 26$ (latitude × longitude × height) tensor and other 22 variables stored as
a $1024 \times 512 \times 221$ (latitude × longitude × time) tensor from 1980/01 to 1998/05. When reading the NetCDF data, a total of
48GB memory will be occupied. The original data we used is double precision, we first process the data into single precision,
and then the existing methods (SZ, ZFP, Blocked-HGFDR) and the proposed method are applied to compare the
compression performances. Research experiments were performed by the MATLAB R2017a environment on a Windows 10
Workstation (HP Compaq Elite 8380 MT) with Intel Corei7-3770 (3.4 GHz) processors and 8 GB of RAM.

The following experiments were performed. (1) In order to transform the original data to data blocks with the balanced
dimension, the dimensions of these data blocks are better to have the same size. Thus, the optimal counts of data blocks
should be determined. For the given compression error, we randomly divide the original data into a series of data blocks with
different block counts, Adaptive-HGFDR is then applied to these data blocks, and the corresponding compression ratios are
calculated. The optimal block count is achieved at the largest compression ratio. (2) Since ESMD have multiple dimensions
and these dimensions may have different organization orders, to verify that the proposed compression method is unrelated
with the data organization order, different variables are selected and organized with different orders. Then the advanced
predict method SZ and the proposed method are applied to these reorganized data to realize the lossy compression, and the
dimensional distributions of compression errors are used to explore the relevance of the method with the data organization
order. (3) To verify the advantages of the proposed method for ESMD, the proposed method was compared with the
advanced transform method ZFP and Blocked-HGFDR. (4) To show the applicability and the aadvantages of the proposed
method for the data with different characteristics, we select 22 variables in ESMD, then the proposed method, ZFP and the
Blocked-HGFDR are applied to compare the compression performances. In these experiments, two key indices are used to
benchmark the performances: the compression error and compression ratio. The compression error is calculated as:
$$\varepsilon = \frac{\left\| T_{Original} - T_{Reconstruction} \right\|^2}{\left\| T_{Original} \right\|^2} \tag{5}$$

Here, the $\| \|^2$ is the F norm. $T_{Original}$ is the original tensor data, $T_{Reconstruction}$ is the compressed tensor data.
The compression ratio $\phi$ is calculated as:
$$\phi = \frac{D_{original}}{D_{compression}} \tag{6}$$
Here, $D_{original}$ is the memory size of original data before compression, $D_{compression}$ is the memory size of the compressed
reconstructed data.
**4.2 Optimal block count selection**
The selection of the optimal block count is carried out using the temperature data (T). Here, the block count with a power of
2 will be the best to fit as the near balanced data blocking. Therefore, a series of block counts of 4, 16, 64, and 128, 256 are
generated as the potential block counts. For the compression constraint, $10^{-4}$ is used as an initial given compression error.
The relationships between the block count (BC) and the compression ratio are shown in Figure 2.
Clearly, the highest compression ratio is reached when the block count equals 16 (BC=16). Hence, the optimum block count
is 16, and the corresponding block size is $256 \times 128 \times 26$. It is interesting to find that the overall compression ratio presents a
downward trend with BC in the range 16 and 64. When BC is larger than 64, the data volume of each block becomes smaller,
and the number of feature components required to achieve the same compression error significantly decrease, so the data
volume of each block after compression significantly decreases. Although the number of blocks is increased (BC=128 and
BC=256), the significant reduction of local block data volume makes the overall compression ratio show an upward trend.
Besides that, the relationship between the block count and the compression ratio is related to the structure and complexity of
the data itself, which is different for the data with different distribution characteristics. For the temperature data (T), the
compression ratio reaches a maximum when the block count is equal to 16.

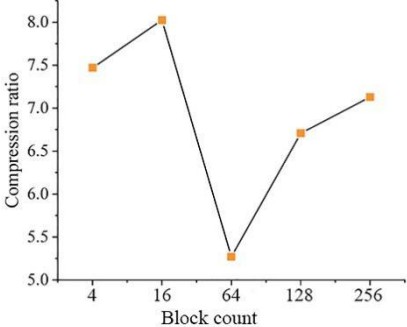

**Figure 2. The relationship between the block count and the compression ratio**

Figure 3 show the original data and the compressed data with different block counts. It can be seen there is no
significant difference between the original data (Figure 3(a)) and the compressed data (Figure 3(b)-Figure 3(f)), and the
distribution characteristics of the compressed data (Figure 3(b)-Figure 3(f)) are consist with the original data (Figure 3(a)).
This may because that the prominent feature components are gradually added to approximate the original data to affect
the compression error, no matter how many blocks are, the proposed method can approach the given compression error
by controlling the rank value to provide the accurate compression results.

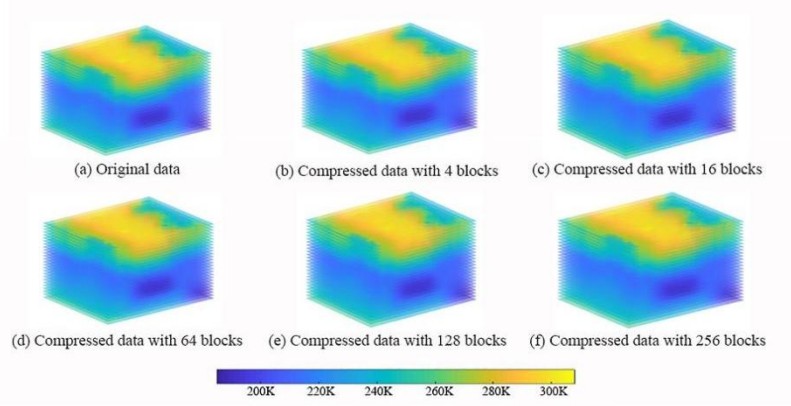

Figure 3. Original data and compressed data with different block counts. (a) The original data; (b) the compressed data when data count is 4; (c) the compressed data when data count is 16; (d) the compressed data when data count is 64; (e) the compressed data when data count is 128;(f) the compressed data when data count is 256.

## 4.3 Comparison with traditional methods

### 4.3.1 Comparison with SZ

In order to verify that the proposed compression method is unrelated with the data organization order, we select three variables $\{\text{SOLIN, TREFMXAV, FSNTC}\} \in \mathbb{R}^{1024 \times 512 \times 221}$ in ESMD. For each variable, we organize the data with different orders as $\{221 \times 512 \times 1024,\ 512 \times 1024 \times 221,\ 1024 \times 512 \times 221\}$. Then, the SZ and the proposed method are applied to the data to realize the lossy compression. The error distributions of different compression results in the corresponding dimension are shown in the Figure 4.

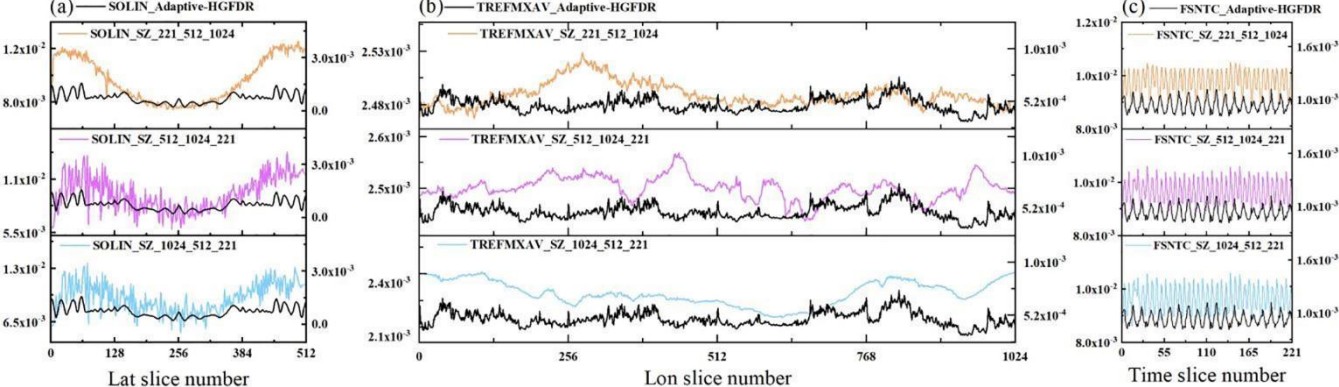


**Figure 4. The compression error distribution along different dimensions. (a) The compression error distribution along latitude for**
**SOLIN. (b) The compression error distribution along latitude for TREFMXAV. (c) The compression error distribution along**
**latitude for FSNTC.**

Figure 4 shows that the dimensional distribution of the compression error in SZ is quite different with the different
organization orders of data. This may because the SZ predicts the data point only along the first dimension but not
along the other dimensions, thus the compression result varies depending on the order of organization. Since the same
ESMD may have the different organization orders, this makes a critical data inconsistency problem of SZ. While,
because the proposed method processes the multidimensional data as a whole, the error distribution is independent
with the data organization order, thus the dimensional distribution of the error remains consistent.
**4.3.2 Comparison with ZFP and Blocked-HGFDR**
To verify the advantage of the proposed method for ESMD, we compare Adaptive-HGFDR with the Blocked-HGFDR and
the ZFP method for the given compression error. Without loss of generality, the relative compression error ratios are set as
$10^{-5}$, $5\times10^{-5}$, $10^{-4}$, $5\times10^{-4}$ and $10^{-3}$ respectively. Here, the block count in the proposed method and the Blocked-HGFDR
method are both set as 16, and the rank of Blocked-HGFDR is selected as the average of the adaptive rank in each divided
block data. In ZFP, the key parameter is the tolerance. For the above given compression errors, we conduct the simulation
experiments with many random tolerances, then find the ideal tolerances in these cases the corresponding compression errors
are close to the given compression errors. Thus, the tolerance parameters are 0.05, 0.3, 0.5, 3.8 and 10. The compression
ratios of different compression methods under the condition of different compression errors are calculated and shown in
Figure 5.

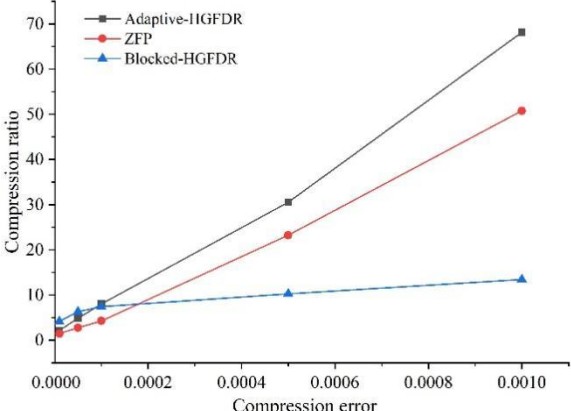

**Figure 5. The relationship between the compression error and compression ratio for different methods.**

Figure 5 shows that as the compression error ratio grows, the compression ratio of all three methods becomes larger and
larger. However, the growth rate of ZFP is much slower than that of Blocked-HGFDR and Adaptive-HGFDR. When the
compression error is less than 0.0001, the compression ratio of ZFP is a little higher than that of Adaptive-HGFDR and
Blocked-HGFDR. This may be because that the approximating of the original data with high accuracy requests higher rank,
which limits the improvement of compression ratio. When the compression error is 0.001, which is also acceptable for most
ESMD data application, the compression ratio of Adaptive-HGFDR increases to 68.16, which means that the compressed
data size is 68.16 times smaller than that of the original data. At the compression error of 0.001, the compression ratio of
Adaptive-HGFDR, ZFP and Blocked-HGFDR are 68.16, 13.42 and 50.78, respectively. The compression ratio of Adaptive-
HGFDR is 5.07 times and 1.34 times larger than that of ZFP and Blocked-HGFDR. These may be because that the Adaptive-
HGFDR can adaptively adjust the compression parameter (rank value) according to the actual data complexity, and thus
better capture data features to improve the compression ratio.
We summarize the error distribution along the longitude dimension of each method in Figure 6. It is clearly seen that the
error distributions of both Adaptive-HGFDR and ZFP are nearly uniform among different longitude dimensions. However,
the Blocked-HGFDR method shows significant four segments of abrupt changes at different longitude slices. The oscillation
characteristics of the three methods are different. For Adaptive-HGFDR, the error distribution is more acted as low-
frequency fluctuations while ZFP method is more as higher frequency fluctuations. The Blocked-HGFDR method has very
different fluctuations characteristics. For the first 1-230 longitude slices, the error distribution of Blocked-HGFDR is of high
frequency fluctuations with relatively high frequency, which is similar to ZFP, while in the rest three segments, it has low
amplitude, which has similar fluctuations as Adaptive-HGFDR. For the comparison of the mean value and standard
deviation of the error distribution among the three methods, the Adaptive-HGFDR has much smaller standard deviation
($6.89 \times 10^{-6}$), compared with ZFP ($2.94 \times 10^{-5}$) and Blocked-HGFDR ($2.80 \times 10^{-5}$). The Blocked-HGFDR method has the smallest
mean compression error ($9.35 \times 10^{-5}$), slightly lower than Adaptive-HGFDR ($9.83 \times 10^{-5}$), while ZFP has the largest mean
compression error ($1.29 \times 10^{-4}$).
Both Blocked-HGFDR and Adaptive-HGFDR show the small difference between the adjacent slices and the big difference
among the different local block data. Due to the spatio-temporal heterogeneity, the feature distributions of each local ESMD
are significantly different, but the feature distributions of adjacent slices have a small difference because of the spatio-
temporal similarity. Meanwhile, since the adjacent compressed slice data have similar characteristics, the error fluctuation of
these slices is small. On the contrary, the structure difference of each compressed local block data is large, and the error
fluctuation is also large. In Blocked-HGFDR, the compression parameter of each block are fixed, and the characteristic
difference of data in each block is ignored. This weakness is improved in Adaptive-HGFDR by adjusting the compression
parameter of each block adaptively according to the compression error to achieve the balanced distribution of error.
Although Blocked-HGFDR performs substantially better for several slice numbers, Adaptive-HGFDR shows less variations.

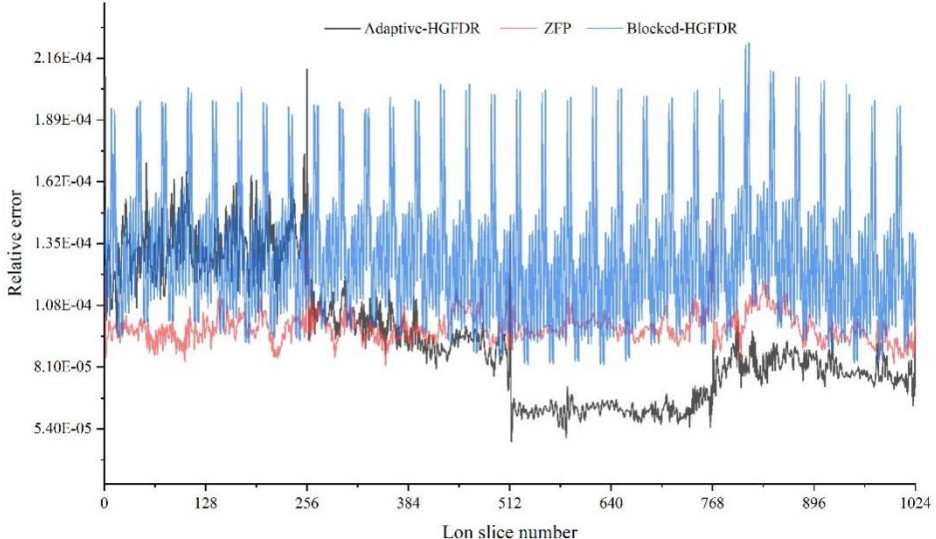

Figure 6. The distributions of compression error along the longitudinal slices ( the slice means the partial data that divided along specific dimensions).

To better reveal the characteristics of the compression error distributions, the distributions of the spatial error for three random spatial pieces (Height 2,8 and 16) are depicted in Figure 7. From Figure 7, we can see that the spatial structure of the data is different at different height, there are both continuous and abrupt structure changes at different levels. Specifically, the compression error in the Blocked-HGFDR method and the ZFP method fluctuates dramatically, forming multiple peaks and valleys. The error distributions of ZFP suggest that there are high frequency stripes. There are irregular spatial patterns for Blocked-HGFDR. The Adaptive-HGFDR method is more stable where the error distribution is nearly random. Additionally, the spatial structure of the data is different at different height, and there are both continuouss and abrupt structure changes at different levels.

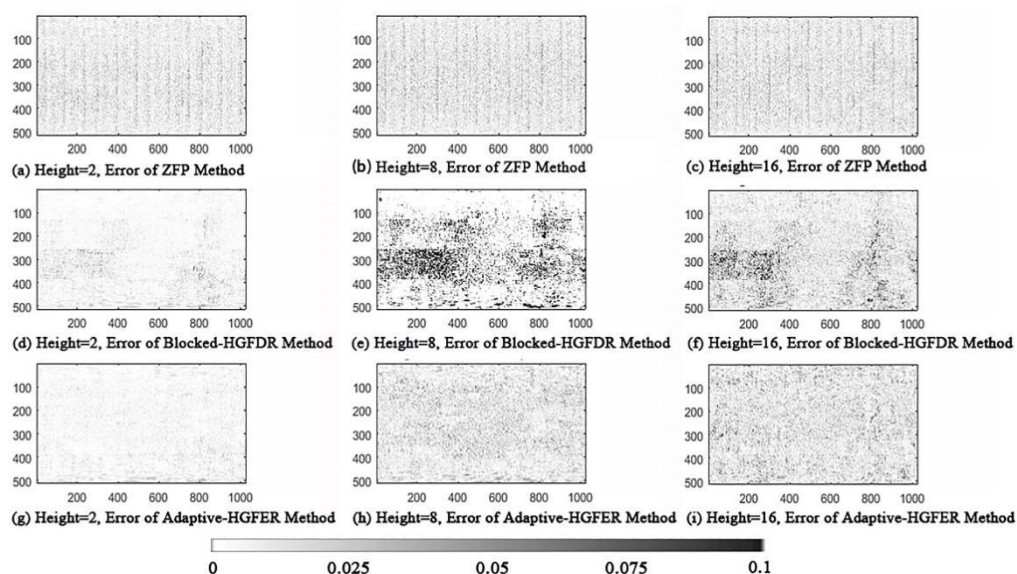

Figure 7. The spatial distribution of compression error of different compression methods. (a)The spatial distribution of compression error with height as 2 in ZFP; (b)the spatial distribution of compression error with height as 8 in ZFP; (c) the spatial distribution of compression error with height as 16 in ZFP; (d) the spatial distribution of compression error with height as 2 in Blocked-HGFDR; (e) the spatial distribution of compression error with height as 8 in Blocked-HGFDR; (f) the spatial distribution of compression error with height as 16 in Blocked-HGFDR; (g) the spatial distribution of compression error with height as 2 in Adaptive-HGFDR; (h) the spatial distribution of compression error with height as 8 in Adaptive-HGFDR; (i) the spatial distribution of compression error with height as 16 in Adaptive-HGFDR;

## 4.4 Evaluation with multiple variables

For a comprehensive comparison of the different methods, 22 monthly climate model data were used as the experimental data. Here, we focus on the variables with flux information and fast changing. Among these variables, there are variables with weak spatio-temporal heterogeneity such as the temperature, and the variables with strong spatio-temporal heterogeneity, which will help to better investigate the applicability of the method. The dimension of the experimental data is $1024 \times 512 \times 221$. Here, considering that the compression error and compression performance of each variable can be comparable, the compression error should not be too big or too small for all the 22 variables, the given error is 0.01, the block size is $256 \times 128 \times 26$, and the block count is 144. For the tolerance parameter settings in ZFP, we conduct the simulation experiments with many random tolerances, then find the ideal tolerances in these cases the corresponding compression errors are close to the given compression errors. A detailed description of the variables is shown in Table 1.

**Table 1: 22 Descriptions of climate model data variables.**

| Variable name | Variable description | Variable name | Variable description |
|---|---|---|---|
| FLDS | Downwelling longwave flux at the surface | PCONVT | Convection top pressure |
| FLDSC | Clearsky downwelling longwave flux at surface | RHREFHT | Reference height relative humidity |
| FLNSC | Clearsky net longwave flux at surface | SOLIN | Solar insolation |
| FLNT | Net longwave flux at top of model | SRFRAD | Net radiative flux at surface |
| FLNTC | Clearsky net longwave flux at top of model | TMQ | Total (vertically integrated) precipitable water |
| FLUT | Upwelling longwave flux at top of model | TREFHT | Reference height temperature |
| FLUTC | Clearsky upwelling longwave flux at top of model | TREFMNAV | Average of TREFHT daily minimum |
| FSDSC | Clearsky downwelling solar flux at surface | TREFMXAV | Average of TREFHT daily maximum |
| FSNSC | Clearsky net solar flux at surface | TS | Surface temperature (radiative) |
| FSNTC | Clearsky net solar flux at top of model | TSMN | Minimum surface temperature over output period |
| FSNTOAC | Clearsky net solar flux at top of atmosphere | TSMX | Maximum surface temperature over output period |


The Adaptive-HGFDR, Blocked-HGFDR, and ZFP method were applied to the 22 variables. The compression ratio, time,
and standard deviation of the slice error were calculated and shown in Figure 8. Form Figure 8(a), it can be seen that
compared with the other two methods, the compression ratio of Adaptive-HGFDR is the largest. This may be because
Adaptive-HGFDR considers the coupling relationship among the spatial-temporal dimensions and searches for the optimal
compression parameter at each data blocks. This not only makes the number of features required by each data block small,
but also makes the effect of data heterogeneity on the compression ratio least. Adaptive-HGFDR captures the data features
more accurate than the other two methods. The adaptive adjustment of parameter makes Adaptive-HGFDR yield the uniform
error distribution for the multiple variables shown in Figure 8(c). In summary, Adaptive-HGFDR provides good adaptability
for ESMD.

Additionally, Figure 8(a) also shows that the tensor-based compression methods (Adaptive-HGFDR, Blocked-HGFDR) have
the high compression ratios for some variables, it may be because for tensor-based compression, the relationship between
data volume and dimensions is transformed from exponential growth to nearly linear growth by defining the tensor product
of tensors, which is essentially the displacement of space by calculating time, so the compression ratio is very high. Also, we
can see that with the given compression error, the compression rates of different variables are significant different. It may be
because different climate model variables have different distribution features. Generally speaking, for the variables with
weak spatio-temporal heterogeneity, a small number of feature components can well achieve the accurate approximation that
have the high compression rate. While, the variables with strong spatio-temporal heterogeneity may need a large number of
feature components that have the low compression rate. Due to the continuous adjustment of compression parameter to
search for the optimal rank, Adaptive-HGFDR is the most time consuming [Figure 8 (b)]. Despite this, some optimization
strategies, such as the spatio-temporal indexes and the unbalanced block split, can help improve the efficiency of Adaptive-
HGFDR.

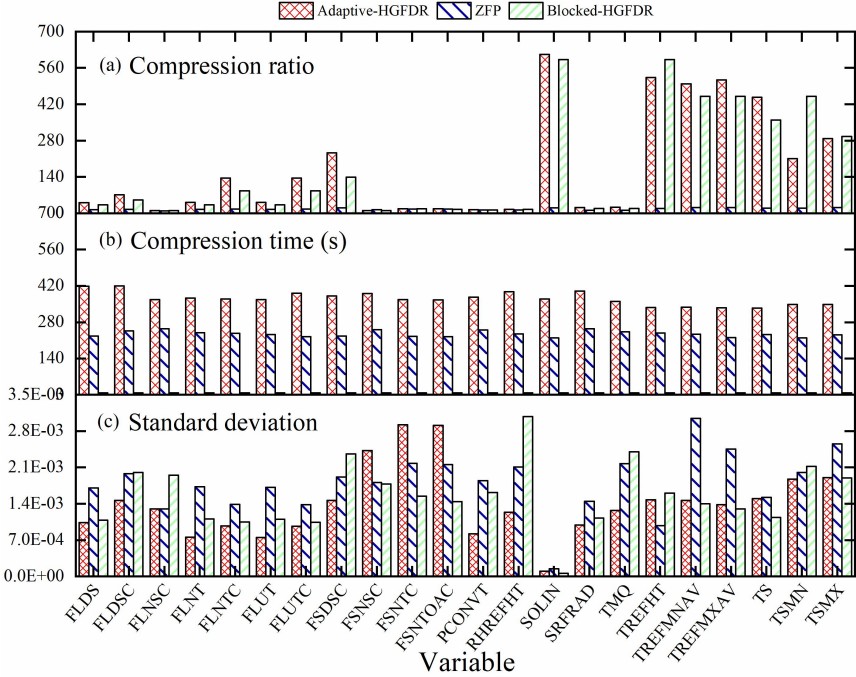


**Figure 8. Comparison results of compression ratio, compression time and standard deviation. (a) The comparison results of**
**compression ratio; (b) The comparison results of compression time; (c) The comparison results of standard deviation.**

## 407 5 Conclusion

In this study, we propose a lossy compression method, Adaptive-HGFDR, for ESMD based on the blocked hierarchical
tensor decomposition by integrating multidimensional coupling correlations. In Adaptive-HGFDR, to achieve the lossy
compression, ESMD is divided into nearly balanced data blocks, which are then approximated by the hierarchical tensor
decomposition. This compression method is applied to all the dimensions of the data blocks rather than mapping the data
into low dimensions to avoid the destruction of coupling correlations among different dimensions. This also avoids the
possible data inconsistency of compression methods like SZ, when the data are extracted and analyzed with different
Input/Output (IO) orders. Thus, this method provides the potential advantage in multidimensional data inspection and
exploration. Additionally, the compression parameter is simple and adaptively calculated for each data block independently
for a given compression error. Therefore, the compression well captures both the global and local variation of the coupling
correlations to improve the approximation accuracy. The simulated experiments demonstrated that, the proposed method has
higher compression ratio and more uniform error distributions than ZFP and Blocked-HGFDR under the same condition, and
can support the lossy compression of ESMD on the ordinary PCs both in terms of the memory occupation and compression
time. Additionally, the comparison results among 22 climate variables show that the proposed method can achieve good
compression performance for the variables with significant spatio-temporal heterogeneity and fast changing.

The application of the hierarchical tensor in this paper provides several new potentials for developing more advanced lossy
compression methods. With the hierarchical tensor, both the representation model and computational model can support the
complex multidimensional computation and analysis(Kressner and Tobler, 2014). For example, commonly used signal
analysis methods like (Singular Value Decomposition)SVD and (Fast Fourier transform)FFT can achieve efficient stream
computing with the hierarchical tensor representation, thus can inherently support efficient on-the-fly computation and
analysis. Other interesting topics focusing on the tensor-based compression, includes the compression for unstructured data
or extremely sparse data (Li, D. et al. 2019). Moreover, comprehensive tensor methods, like Partial Differential Equation
(PDE) are also recently been introduced to the hierarchical tensor, Thus, it is even possible to integrate some dynamic
models of earth systems directly on the compressed data. With the rapid development of the tensor theory and applications, it
may provide more and more potentials for tensor-based spatio-temporal data compression for the modelling and analyzing of
ESMD.

Multiple dimensionality and heterogeneity are the natural attributes of ESMD. In ESMD, there are various spatio-temporal
structures with gradual/sudden change and fast/slow change, which also show the significant regularity and randomness.
From the perspective of the rules of ESMD distribution, constructing the data compression method based on
multidimensional coupling correlations may be the key to improve ESMD compression performance in the future. For
example, for static or slow-varying variables, large block and small Rank can be used to achieve large compression, while
for fast-changing variables, small block and large Rank may be needed. The data coupling correlations obtained by
dynamically adjusting the block count and Rank, can not only be used to the data compression, but also are helpful to realize
the data organization and compressed storage based on the data characteristics. Additionally, in the large-scale simulation
experiment with long time sequence and multi-mode integration, this characteristic-based data organization and storage of
multidimensional ESMD make it possible to only retain the prominent components, so as to achieve efficient comparison of
large-scale data and can help to promote the ability of ESMD application service. For instance, for the major natural
disasters, this multidimensional tensor compression can support the progressive transmission with the limited bandwidth by
using only the prominent components, which can help to promote the depth and breadth of ESMD application.
**Code and data availability.** The Adaptive-HGFDR lossy compression algorithm proposed in this paper was conducted out
in MATLAB R2017a. The exact version of Adaptive-HGFDR and experimental data used in this paper is archived on
Zenodo(AndyWZJ, 2020). The experimental data are Large-scale Data Analysis and Visualization Symposium Data

obtained from (OSDC) Open Science Data Cloud. This data set consists of files from a series of global climate dynamics simulations run on the Titan supercomputer at Oak Ridge National Laboratory in 2013 by postdoctoral researcher Abigail Gaddis, Ph.D. The simulations were performed at approximately 1/3-degree spatial resolution, or a mesh size of 1024x512 for 2D. We downloaded this simulation data in the common NetCDF (network Common Data Form) format in 2016 from https://www.opensciencedatacloud.org/. The code of the all algorithms and comparative test are provided and can be download form http://doi.org/10.5281/zenodo.4384627.

**Author contribution.** Zhaoyuan Yu, Linwang Yuan and Wen Luo designed the paper's ideas and methods. Zhengfang Zhang and Yuan Liu implemented the method of the paper with code. Zhaoyuan Yu, Zhengfang Zhang and Dongshuang Li wrote the paper with considerable input from Linwang Yuan. Zengjie Wang revised and checked the language of the paper.

**Funding.** This work was financially supported by the National Natural Science Foundation of China[41625004 41971404] and the National Key R&D Program of China[2017YFB0503500].

**Competing interests.** The authors declare that they have no conflict of interest.

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
