# Peer review of "Lossy compression of earth system model data based on hierarchical"

_Geoscientific Model Development, 2020_

## Referee Comment (RC1) · Anonymous Referee #1 · 1 Jul 2020

The authors describe Adaptive-HGFDR, a lossy compression method that adapts to the data such that error distributions are uniform. They apply the method to several atmospheric variables. The writing in the manuscript needs significant improvement. Also many details seem to have been left out (addressed in comments below), leaving the reader with many questions.

————————————— Comments/suggestions/questions —————————————

-This manuscript contains many grammar issues and needs to be written better. Many phrases are awkward or do not make sense.

-Abstract: does "stable compression error" mean uniform distribution or error? This is
odd terminology. (Also in line 155 and 231)

-line 26: Not all ESMD data is high-precision. In fact, it is typical that calculations are done in double precision, but that data is output in single precision (e.g., for CESM). What is the precision of the data that you are compressing?

- line 30-32: "lossless compression has an upper limit of compression ratios" - I'm not clear what is meant by this. It really depends on the data so I don't know how you'd define an upper limit. Loseless compression works quite well on smooth data, for example. Also I don't get "which grows much slower than the velocity of data volume grows". Is the upper limit growing? What is meant by the velocity of the data volume?

-line 38: I'm not sure that I agree with "the error cannot be controlled in a data-driven way". I would say that the SZ method in Liang at al 2019 ("Significantly Improving Lossy Compression Quality Based on an Optimized Hybrid Prediction Model") is data driven.

-line 39: what is meant by "the data compression parameters of common data files are relatively fixed". Different variables within a file could be compressed different amounts.

-the distinction between file-based and encoding-based compression is not clear to me. Are they meant to be mutually exclusive? Why can't one use an encoding based method on the entire file. Please explain this more clearly.

- the references need to be carefully checked. There are a lot of errors! Many are incomplete due to missing information (page numbers, journal names, ...) Also I noticed incomplete/incorrect author lists, multiple entries for the same article, ...

-line 60: "intercepting the floating-point precision" does not make sense

-lines 61-62: "As the distribution of floating-point precision of data is not uniform, the compression errors may also distribute unevenly. Therefore, it is difficult to control the distribution of data compression error." This statement needs to be clarified (it is not true for all data).

-line 66: does zfp really use "feature prediction -based" encoding? It is a transform method - is that what you mean? Please clarify what is meant by "feature prediction-based encoding" ...

-Intro: consider including SZ compression - it's very popular

-line 74: "values in neighboring ranges tend to be numerically close to each other" This is not true for all data in ESMD - a number of variables have abrupt changes (e.g., clouds).

-section 2: "the partial block data is not only much smaller than the original geographic spatial tensor"- you need to explain what is meant by the partial block data. The discussion just starts talking about blocks, which can mean a lot of different things, and we don't want to assume the reader is familiar with previous work.

-Figure 1: "Judging the rank meets the error" is awkward...

-Section 2 was not that useful in terms of getting the big picture. It should be written as a more general overview as title implies (Basic Overview) - so avoid (or define) undefined terms (e.g., the fast search method, block of divided data, target error, ...) Consider beginning with explaining the spatial tensor as in the flow chart.

-line 139: what is meant by making the blocks an "ideal size"?

-Please motivate why uniform error distribution is important? In some cases, there may be a need for more accuracy in some regions than others.

-line 163: It's not clear what alpha and beta are...

-Section 4.1.: what model is this data from? More specific info on the data is needed (or refer to the section at the end), and in line 189: this reference does not make sense for NetCDF: Springer, U. S.: Community Earth System Model (CESM), Encycl. Parallel Comput., 351, 2011. (Also I'm not sure what this is referring to). Even in the data availability section is doesn't say what model was used. Also is there a doi for the data

or how do I find the data on Data Cloud? (Now I see that this is CESM data - it's only written in the abstract. Also why the choice of data from 2013?)

-Figure 4; can you better explain how the block size is affecting the compression ratio (resulting in this v-shape)? Is this behavior "typical" or expected?

-section 4.2 - how does the block number relate to the block size? please clarify the distinction.

-section 4.2 - is 1e-4 relative error or absolute error? Also is this a max error or an average error (e.g., rmse)?

-Figure 3: The caption is on a different page than the figure. Also it is hard to see what is going on here. Consider plotting the errors instead. (I assume you are plotting temperature in K, though it doesn't say that).

-section 4.3: How is zfp being applied to the data? It's effectiveness quite depends on the spatial locality. Also why did you choose zfp?

-section 4.3: Why would the compression ratio of the HGFDR go up so much with a change in error tolerance from 1e-4 to 5e-4? This needs to be explained as it is hard to believe.

-I don't quite understand what is being plotted in figure 6. But the regular pattern in zfp error is likely explained by its block size. (e.g. Hammerling et al 2019 "A Collaborative Effort to Improve Lossy Compression Methods for Climate Data"). It would be helpful to discuss this. ZFP block patterns are also evident in figure 7a.

-section 4.4: Why did you pick .01 for the error limit? Is this a relative error? Errors limits were smaller in the previous section.

-section 4.4: Why did you pick those particular variables? Do they have a good rep-resentation of the different types? For example, temperature variables are "easy" to compress as compared to variables with discontinuities and large dynamic ranges (like

precipitation). So having a variety of test variables is important. There are hundreds of variables in the CESM atmospheric component.

-line 249: Please clarify: "maintains the maximum compression ratio under the constraints of the same compression error"

-line 251: "removes data redundancy" - please clarify what you mean (all compression methods are trying to model the data so as to remove redundancy).

-section 4.4: More discussion is needed to explain why the compression ratios vary so much in figure 8. Some of these CRs are very high and I question what the error looks like. The .01 threshold is large, but a 600x reduction is pretty shocking really. Also instead of sharing the std dev - shouldn't we see some sort of average or max error instead?

———————————- Minor items: ———————————-

- "Earth" should be capitalized everywhere in "Earth system model"

- the ZFP method should be referenced by Lindstrom 2014 (line 65)

- line 88, 97: What is the citation for the first HGFDR work? (I think it's Yuan 2015, but you have not made that clear here.)

-line 171: compression is misspelled....

---

## Referee Comment (RC2) · Anonymous Referee #2 · 27 Jul 2020

Summary:

The authors present a tensor-based lossy compression method that is based on HGFDR, which compressed netCDF files across all variables, rather than a single variable at a time which many other methods use. They cite most of the important references and compare their method to some state-of-the art methods. The main idea of this work is a good one and in parts, it's described in sufficient technical depth. However, the authors assume too much prior knowledge of the basics of this techniques, which should be introduced and defined clearly. This technique produced promising results, but the analysis of the results lacks some depth. Overall, the paper definitely

needs some work on clarity of writing, formatting, wording, and typos.

_____ Detailed comments:

Missing definitions: - line 106: Explain the concept behind block data/partial block data. This is not common knowledge and it is an important basis of your method. - line 121: Explain what you mean by dichotomy (a simple definition will do for this one) - Section 4: define terms you use. What do you mean by slice, height, block number?

Clarify in writing: - Lines 23-27 are phrased in a somewhat confusing way (especially step 1). Turning this into a list may be helpful, and the Figure could be tied in more efficiently by referring to specific part of the flow chart. - Most of the Figures have insufficient captions. Please provide enough caption that the take-away message of each Figure is clear. E.g. which method performs best? Why should the reader care? - Figure 1: This flow chart seems incomplete. The iteration over different versions of the compression method is not represented adequately (e.g. there are multiple iterations until an optimal rank is found but the chart implies it's a single step) - lines 175-180: the list format is good but the style is inconsistent. Consider leading each row with a verb, and provide a natural language description of the steps which only have a formula. - line 181: O(log n) is claimed but is missing a justification (or proof). - line 189: If you only provide a single variable, put in a reference to Table 1. Furthermore, please explain why you chose this variable. A better way to put this may be to introduce the data as "this is a tensor with 23 attributes (full list later on in Table 1)." - line 191-192: what's the average memory occupancy/usage (not occupation!) of each variable? This would be much more valuable information than that of a single variable. - Section 4.2 and later: consider using "block count" instead of "block number", as the latter could also be a number (index) assigned to a single block. - line 204: these numbers are very likely not random... instead of claiming randomness, it would be better to provide a reason for these choices. Are you trying to look at different orders of magnitude? Furthermore, 256 is missing here although it is present in the picture and later descriptions - Figure 3/line 213: this may also be due to the colormap as the one chosen for this
[Figure]

Figure has very little color depth. The "hot" colormap in the same color family would provide better differentiation between values. The "viridis" colormap would be another good choice. - Figure 5: what type of error is this? Also, the scale on the x-axis is very hard to read. A side by side comparison with a more consistent scale may be helpful. Furthermore, readers who are not familiar with compression will appreciate some guidance on reading compression vs error charts. - line 228/229: This sentence does not make sense - line 228-230: This is not enough analysis for this chart. - line 232: again, not random. Why these numbers and not different ones? - Figure 6/analysis: blocked-HGFDR performs substantially better for a lot of slice numbers and despite some bigger changes, the error seems largely consistent for adjacent slices. Why is that? And since this method builds on blocked-HGFDR, why does this not happen for adaptive HGFDR? - Figure 6: Are those the numbers of the slices, or the count of slices overall? - Figure 7: What is "height"? Is that the number of layers? variables? What are the differences between different heights? How are you sub-selecting these layers/variables? - Figure 7: What type of error is this? - Figure 7: Color may be helpful. - line 243: "error limit" should probably be threshold"? Which type of error are you looking at? - Figure 8: insufficient discussion of compression time – why is this algorithm so much slower than the competition? Especially so much slower that Blocked-HGFDR which barely shows up on the chart and which provides similar compression ratios

Open questions: - Can this method work in situ with the simulation? Why/why not? - Under which conditions does the proposed algorithm perform better? Under which conditions are different algorithms better? - Can this algorithm be applied to other types of data? Why/why not?

References: - Overall, the authors provide a good overview of general compression work as well as some more domain-specific examples. - However, the formatting of these references needs some work. There is a lot of information that's missing, including author names (Anon, "Of and Acm", "None", "Matrices",) - I'm not sure if all references are summarized and categorized appropriately in the text. They're grouped

a bit weirdly

Minor issues: - various references are missing a space before the parenthesis (line 25, 28, and a couple of others) - line 53: vector quantization(VQ) (Vector Quantization) – remove duplicate - line 69: performance of the compression may *be* low - line 133: "dimenisonal" should be "dimensional" - line 137: missing space before Thus - Eq 2 and several other places: weird formatting of "res" – please use consistent font type - line 162: align alpha and beta with text - line 170: "conpression" should be "compression - line 172 and other places: remove space in "R_M ax" and "R_M in" - line 193: "in a MATLAB environment"? I would assume they were performed *by* the authors. - Figure 3: caption is not on the same page as the Figure. - Figure 6: "longitudanal" should be "longitudinal" - line 243: dimension should probably be resolution - Table 1: the alignment here is very confusing. Variable names are aligned to the bottom and variable descriptions are aligned to the top. This is very awkward to read. - line 253: replace square brackets with parentheses - line 254: "We pay more attention" should probably read "The main focus of this work"... - line 265: Missing "The" at the beginning of the sentence - line 266: "maintain" should be "maintains"

―――――――――――――――

---

## Author Comment (AC1) · 26 Aug 2020

Dear Editor and Reviewers:

This is a reversion of our former manuscript gmd-2020-124. Thank you for your interest and helpful comments on our paper. In the revised version, we reorganized our contents, added several important technological details, and extended the experiments and evaluations. To improve the language expressions, we have carefully checked and modified the manuscript accordingly, and hope this time our paper will meet the high standard criteria of the Geoscientific Model Development.

[Figure]

Detailed responses to the comments are listed as follow:

Reviewer #1: The authors describe Adaptive-HGFDR, a lossy compression method that adapts to the data such that error distributions are uniform. They apply the method to several atmospheric variables. The writing in the manuscript needs significant improvement. Also many details seem to have been left out (addressed in comments below), leaving the reader with many questions.

(1) This manuscript contains many grammar issues and needs to be written better. Many phrases are awkward or do not make sense.

The manuscript has been revised carefully.

(2) Abstract: does "stable compression error" mean uniform distribution or error? This is odd terminology. (Also in line 155 and 231)

"Stable compression error" means the uniform distribution of compression error, we have corrected the corressponding expression.

(3) line 26: Not all ESMD data is high-precision. In fact, it is typical that calculations are done in double precision, but that data is output in single precision (e.g., for CESM). What is the precision of the data that you are compressing?

Yes. The original data we used is double precision, we first process the data into single precision, and then compress it with the proposed method.

(4) line 30-32: "lossless compression has an upper limit of compression ratios" - I'm not clear what is meant by this. It really depends on the data so I don't know how you'd define an upper limit. Loseless compression works quite well on smooth data, for example. Also I don't get "which grows much slower than the velocity of data volume grows". Is the upper limit growing? What is meant by the velocity of the data volume?

We have modified the corresponding expression into "Which grows much slower than the velocity of data volume grows" refers to the increasing amount of data that needs to

be processed, while the research on lossless compression is progressing slowly. We have added the corresponding explanations in page 1 line 30∼31. (5) line 38: I'm not sure that I agree with "the error cannot be controlled in a data-driven way". I would say that the SZ method in Liang at al 2019 ("Significantly Improving Lossy Compression Quality Based on an Optimized Hybrid Prediction Model") is data driven. line 39: what is meant by "the data compression parameters of common data files are relatively fixed". Different variables within a file could be compressed different amounts.

We have delected the expression"the error cannot be controlled in a data-driven way". "the data compression parameters of common data files are relatively fixed" means the file-based compression methods cannot arbitrarily adjust the compression parameter according to the any given compression error. We have modified the corresponding expression in page 2 line 54∼57.

(6) the distinction between file-based and encoding-based compression is not clear to me. Are they meant to be mutually exclusive? Why can't one use an encoding based method on the entire file. Please explain this more clearly.

We have revised the review of existing methods in introduction part. Considering that the main idea of ESMD lossy compressions is to eliminate unnecessary or redundant information in data to reduce the data size. There are two different kinds of information that can be considered as unnecessary or redundant in ESMD: information of data descriptions and information of data features. Therefore, there are two major different types of ESMD lossy compression methods: the description-based lossy compression and feature-based lossy compression.

(7) the references need to be carefully checked. There are a lot of errors! Many are incomplete due to missing information (page numbers, journal names, ...) Also I noticed incomplete/incorrect author lists, multiple entries for the same article, ...

All references have been carefully checked, the incomplete information has been added and the incorrect information has been corrected.

(8) line 60: "intercepting the floating-point precision" does not make sense

We have modified the coressponding expression as " implements the data compression through controlling the precision of a floating-point expression of the original data, and eliminating redundant floating-point precision", in page 2 line 49~50.

(9) lines 61-62: "As the distribution of floating-point precision of data is not uniform, the compression errors may also distribute unevenly. Therefore, it is difficult to control the distribution of data compression error." This statement needs to be clarified (it is not true for all data).

We have modified the corresponding expression as "for the error truncation-based compression, the distribution of floating-point precision of ESMD is not uniform, which could lead to the unevenly distribution of compression errors" in page 2 line 55~57.

(10). line 66: does zfp really use "feature prediction -based" encoding? It is a transform method - is that what you mean? Please clarify what is meant by "feature prediction-based encoding" ...

For the feature prediction-based method, it use parametric functions to fit the data and predict the structure of data. Then the function parameters are used to represent the original data in a compact form, reducing the data volume. We have modified the corressponding expression in page 3 line 73~76. ZFP are typical methods that use the feature prediction to achieve lossy compression.

(11). line 74: "values in neighboring ranges tend to be numerically close to each other" This is not true for all data in ESMD - a number of variables have abrupt changes (e.g., clouds).

We have modified the corresponding the expression as " For some ESMD variables like temperature, solar and longwave flux, there are significant correlations between different dimensions, i.e., values in neighboring ranges tend to be numerically close to each other." in page 3 line 84~86.

(12). -section 2: "the partial block data is not only much smaller than the original geographic spatial tensor"- you need to explain what is meant by the partial block data. The discussion just starts talking about blocks, which can mean a lot of different things, and we don't want to assume the reader is familiar with previous work.

For the spatio-temporal data, it can be seen that it is composed of a series of local data with the same spatio-temporal reference. These local data are defined as the block data. We have added the coressponding defintion and explantion in page 6 line 150∼157.

(13). Figure 1: "Judging the rank meets the error" is awkward...

We have reproduced the Figure 1.

(14). -Section 2 was not that useful in terms of getting the big picture. It should be written as a more general overview as title implies (Basic Overview) - so avoid (or define) undefined terms (e.g., the fast search method, block of divided data, target error, ...) Consider beginning with explaining the spatial tensor as in the flow chart.

The section 2 has been revised, and added the corresponding definition in section 3.1.

(15). -line 139: what is meant by making the blocks an "ideal size"?

Considering the dimensional imbalance of ESMD data, for example, the data in the temporal dimension is typically longer than that in the spatial dimension for a spatio-temporal series with long observations. This dimensional imbalance not only enlarges the overall fitting error during the data fitting process for tensor decompression, but also makes it difficult to achieve fine control of the compression ratio and error distribution. Therefore, it is better to split the original data into small local data blocks, achieving a more balanced dimension structure in each local data block. So the ideal size menas the relativly balanced dimension structure. We have modified the corressponding expression in page 4 line 119∼124.

(16). -Please motivate why uniform error distribution is important? In some cases,

**[GMDD](https://gmdd.net)**
there may be a need for more accuracy in some regions than others.

For the ESMD compression, the reduction of data accuracy may lead to the deviation in calculating the long-term trend, change rate, inflection point and abrupt change, and further affect the data quality as well as the subsequent analysis of ESMD, especially for the variables with strong spatial heterogeneity like temperature or fluxes. The heterogeneity of original data distribution could lead to the uneven distribution of compression errors and make it difficult to separate the compression error from the spatial-temporal characteristics of original ESMD. Therefore, keeping the distribution of compression error as uniform as possible can improve the data quality of ESMD lossy compression. We have added the coressponding explanation in page 2 line 28~39.

(17). -line 163: It's not clear what alpha and beta are...

Equation (3) is mainly used to demonstrate the relationship between the compression parameter and compression error, for the given compression error, compression parameter can be adjusted according to the equation (3) to maintain the stable distribution of compression error in each local data. In equation (3), are the coefficients depended on the structure and complexity of the data, which can be obtained by the simulation experiment for actual data. We have corrected the corresponding expression in page 7 line 182~183.

(18). -Section 4.1.: what model is this data from? More specific info on the data is needed (or refer to the section at the end), and in line 189: this reference does not make sense for NetCDF: Springer, U. S.: Community Earth System Model (CESM), Encycl. Parallel Comput., 351, 2011. (Also I'm not sure what this is referring to). Even in the data availability section is doesn't say what model was used. Also is there a doi for the data or how do I find the data on Data Cloud? (Now I see that this is CESM data - it's only written in the abstract. Also why the choice of data from 2013?)

We have delected the reference in line 189. In this paper, data produced by Community Earth System Model are used as experimental data to evaluate the compression performance of Adaptive-HGFDR, which can be obtained from Open Science Data Cloud in NetCDF (Network Common Data Form) format(http://doi.org/10.5281/zenodo.3997216). We have added the coressponding explantion in page 8 line 213~217.

(19). Figure 4; can you better explain how the block size is affecting the compression ratio (resulting in this v-shape)? Is this behavior "typical" or expected?

The highest compression ratio is reached when block counts equals to 16 (BC=16). Hence, the optimum block count is 16, and the corresponding block size is . Interesting things can be revealed that the overall compression ratio presents a downward trend with BC in the range 16 and 64. When BC is larger than 64, the data volume of each block becomes smaller, and the number of feature components required to achieve the same compression error significantly decrease, so the data volume of each block after compression significantly decreases. Although the number of blocks is increasing (BC=128 and BC=256), the significant reduction of local block data volume makes the overall compression ratio show an upward trend. Besides that, the relationship between the block counts and compression ratio is related to the structure and complexity of the data itself, which is different for the data with different distribution characteristics. We have added the corresponding explanations in page 9 and 10 line 243~256.

(20). -section 4.2 - how does the block number relate to the block size? please clarify the distinction.

For the given data, it is divided into a series of data block with the same data size. Here the data size refer to the size of each dimension of multidimensional data, and the block count (block number) is determined by the ratio between size of divided data block and that of original data. And the block counts will affect the data fitting performance, the compression ratio and the complexity of file system I/O (input/output). More blocks may achieve smaller rank for accurate fit of each block and finer control of the error distribution of overall compression but may lead to more parameters and file system

[Figure]

I/O. On the contrary, if the block counts is small, the heterogeneity in one data block may be large, which will result in a low accuracy of data fitting and may require higher rank for the compression. Therefore, selection an optimal block counts to meet the largest compression ratio. We have added the corresponding explanations in page 9 line 235~242.

(21). -section 4.2 - is 1e-4 relative error or absolute error? Also is this a max error or an average error (e.g., rmse)?

The compression error in this manuscript refer to the relative error ratio, we have added the coressponding explantion in page 8 Equation (4).

(22). -Figure 3: The caption is on a different page than the figure. Also it is hard to see what is going on here. Consider plotting the errors instead. (I assume you are plotting temperature in K, though it doesn't say that).

We have reproduced the Figure 3.

(23). -section 4.3: How is zfp being applied to the data? It's effectiveness quite depends on the spatial locality. Also why did you choose zfp? We apply zfp to each spatial slice of the data. For the overall data with a size of 1024X512X26, we first split the data into 26 spatial slice and compress each slice with zfp. The reason we choose zfp is zfp have public source code and baseline benchmark. Therefore, the implementation affection to the compression performance can be reduced.

(24). -section 4.3: Why would the compression ratio of the HGFDR go up so much with a change in error tolerance from 1e-4 to 5e-4? This needs to be explained as it is hard to believe. This may because in the hierarchical tensor decomposition, the tensor approximation error is a power exponent relation with the parameter.

(25). -I don't quite understand what is being plotted in figure 6. But the regular pattern in zfp error is likely explained by its block size. (e.g. Hammerling et al 2019 "A Collaborative Effort to Improve Lossy Compression Methods for Climate Data"). It would be

helpful to discuss this. ZFP block patterns are also evident in figure 7a.

We agree that the error structure may related to the block size of zfp. As pointed by Hammerling et al 2019, the mechanism of zfp determined that the compression error affected by the block size seems can not be avoid. And the only modifiable parameter of zfp is the tolerance, which controls the accuracy of the compression. We modified the discussion section, added the detailed parameters we used for zfp, discussed the possible error pattern that may be caused by block size of ZFP in page 12 line 314∼319.

(26). -section 4.4: Why did you pick .01 for the error limit? Is this a relative error? Errors limits were smaller in the previous section.

We pick .01 for the error limit for two reasons: 1. We hope that the compression error and compression performance of each variable can be comparable, which means the error is not too big or too small for all the 22 variables, even the numerical values of the original 22 variables are very different. When we pick error limit of .01, the evaluation data can be clearly represented in a single graph. 2. We hope to know the maximum potential that how much space we can saved for the ESMD. As we have revealed the that for the error limit of 0.001, the improvement of our method and Zfp is not that significant. When the error limit relaxed a little, the performance of our method will greatly improve. As you can seen in Figure 6, the compression ratio can be as high as 400 ∼ 600 for the some variables. We have dded the corresponding explantion in page 13 line 328∼330.

(27). -section 4.4: Why did you pick those particular variables? Do they have a good representation of the different types? For example, temperature variables are "easy" to compress as compared to variables with discontinuities and large dynamic ranges (like precipitation). So having a variety of test variables is important. There are hundreds of variables in the CESM atmospheric component.

In this manuscript, we focus on the variables with flux information and fast changing.

Among these variables, there are variables with weak spatio-temporal heterogeneity such as the temperature, and the variables with strong spatio-temporal heterogeneity, which will help to better demonstrate the applicability of the method. We have added the corressponding explantion in page 13 line 324∼327.

(28). -line 249: Please clarify: "maintains the maximum compression ratio under the constraints of the same compression error"

For the given compression error, compared with the another two methods, the compression ratio of the proposed method is the larggest when approach the given compression error. We have corrected the coressponding expression in page 14 line 333∼334.

(29). -line 251: "removes data redundancy" - please clarify what you mean (all compression methods are trying to model the data so as to remove redundancy).

Because Adaptive-HGFDR considers the coupling relationship among the spatial-temporal dimensions and search for optimal compression parameters at each data blocks. This not only makes the number of features required by each data block is small, but also makes the effect of data heterogeneity on the compression ratio least. Adaptive-HGFDR captures the data features more accurate than the other two methods. We have corrected the corresponding expression in page 14 line 334∼343.

(30). -section 4.4: More discussion is needed to explain why the compression ratios vary so much in figure 8. Some of these CRs are very high and I question what the error looks like. The .01 threshold is large, but a 600x reduction is pretty shocking really. Also instead of sharing the std dev - shouldn't we see some sort of average or max error instead?

We have added the mean compression error in page 12 line 297. Because different climate model variables have different distribution features, for the given compression error, the compression rates of different variables are significant different. Generally,

for the variables with weak spatio-temporal heterogeneity, a small number of feature components can well achieve the accurate approximation that have the high compression rate. While, the variables with strong spatio-temporal heterogeneity may need a large number of feature components that have the low compression rate. Additionally, for tensor-based compression, the relationship between data volume and dimensions is transformed from exponential growth to nearly linear growth by defining the tensor product of tensors, which is essentially the displacement of space by calculating time. So the compression ratio of compression is very high. We have added the corresponding explanations in page 14 line 340~348.

Reviewer #2: The authors present a tensor-based lossy compression method that is based on HGFDR, which compressed netCDF files across all variables, rather than a single variable at a time which many other methods use. They cite most of the important references and compare their method to some state-of-the art methods. The main idea of this work is a good one and in parts, it's described in sufficient technical depth. However, the authors assume too much prior knowledge of the basics of this techniques, which should be introduced and defined clearly. This technique produced promising results, but the analysis of the results lacks some depth. Overall, the paper definitely needs some work on clarity of writing, formatting, wording, and typos.

Our concerns of this paper:

(1). Missing definitions: - line 106: Explain the concept behind block data/partial block data. This is not common knowledge and it is an important basis of your method. - line 121: Explain what you mean by dichotomy (a simple definition will do for this one) - Section 4: define terms you use. What do you mean by slice, height, block number?

We have added the coressponding defintions and explantions in page 5 line 150~157, page 7 and page 8 line 203~210, page 12 line 310~311.

(2). - Lines 23-27 are phrased in a somewhat confusing way (especially step 1). Turning this into a list may be helpful, and the Figure could be tied in more efficiently by

referring to specific part of the flow chart. Most of the Figures have insufficient captions. Please provide enough caption that the take-away message of each Figure is clear. E.g. which method performs best? Why should the reader care?

We have reproduced all Figures and added the corressponding explantion.

(3). Figure 1: This flow chart seems incomplete. The iteration over different versions of the compression method is not represented adequately (e.g. there are multiple iterations until an optimal rank is found but the chart implies it's a single step)

We have reproduced the Figure 1, and added the corressponding explantion in the page

(4). - lines 175-180: the list format is good but the style is inconsistent. Consider leading each row with a verb, and provide a natural language description of the steps which only have a formula.

We have added the corresponding explanations in section 2.

(5). line 181: O(log n) is claimed but is missing a justification (or proof). We have modified the expression of algorithm to make it much clearer in page 8 line 207∼210. We added the reference of the complexity.

(6). - line 189: If you only provide a single variable, put in a reference to Table 1. Furthermore, please explain why you chose this variable. A better way to put this may be to introduce the data as "this is a tensor with 23 attributes (full list later on in Table 1)."

In this manuscript, we focus on the variables with flux information and fast changing. Among these variables, there are variables with weak spatio-temporal heterogeneity such as the temperature, and the variables with strong spatio-temporal heterogeneity such as the precipitation, which will help to better demonstrate the applicability of the method. We have added the corresponding explanations in page 13 line 323∼326.

(7). - line 191-192: what's the average memory occupancy/usage (not occupation!) of each variable? This would be much more valuable information than that of a single variable.

The data set includes air temperature data (T) stored as a (latitude longitude height) tensor and 22 other attributes stored as a (latitude longitude time) tensor from 1980/01 to 1998/05. When reading the NetCDF data, a total of 48GB memory will be occupied. We have added the corresponding explanations in page 8 line 216~217. (8). - Section 4.2 and later: consider using "block count" instead of "block number", as the latter could also be a number (index) assigned to a single block.

We have modified the corresponding expression in section 4.2.

(9). - line 204: these numbers are very likely not random... instead of claiming randomness, it would be better to provide a reason for these choices. Are you trying to look at different orders of magnitude? Furthermore, 256 is missing here although it is present in the picture and later descriptions - Figure 3/line 213: this may also be due to the colormap as the one chosen for this Figure has very little color depth. The "hot" colormap in the same color family would provide better differentiation between values. The "viridis" colormap would be another good choice.

Generally, the data for each block is better to have the same size. And the block counts with a power of 2 will be best to fit as the near balanced data blocking. Therefore, a series of block counts of 4, 16, 64, and 128, 256 are generated as potential block counts. Secondly, the given compression error may affect the optimal block count, we will setup an initial given compression error of 10-4 to conduct the experiments. We have added the corresponding explanations in page 9 line 243~246 and reproduced the Figure 3.

(10). - Figure 5: what type of error is this? Also, the scale on the x-axis is very hard to read. A side by side comparison with a more consistent scale may be helpful. Furthermore, readers who are not familiar with compression will appreciate some guidance on

reading compression vs error charts.

The error is the relative error ratio, we have added the corresponding definition in equation 4, and reproduced Figure 5 with the consistent scale.

(11). - line 228/229: This sentence does not make sense, this is not enough analysis for this chart.

We have modified the corresponding expression in page 11 line 273∼274.

(12). - line 232: again, not random. Why these numbers and not different ones?

10-4 is juat set as an initial given compression error of to conduct the experiments in this work.

(13). - Figure 6/analysis: blocked-HGFDR performs substantially better for a lot of slice numbers and despite some bigger changes, the error seems largely consistent for adjacent slices. Why is that? And since this method builds on blocked-HGFDR, why does this not happen for adaptive HGFDR?

Both Blocked-HGFDR and Adaptive-HGFDR show the small difference between the adjacent slices and the big difference among the different local block data. Due to the spatio-temporal heterogeneity, the feature distributions of each local ESMD are significantly different, but the feature distributions of adjacent slices have a small difference because of the spatio-temporal similarity. Meanwhile, since the adjacent compressed slice data have similar characteristics, the error fluctuation of these slices is small. On the contrary, the structure difference of each compressed local block data is large, the error fluctuation is also large. In Blocked-HGFDR, the compression parameters of each block are fixed, and the characteristic difference of data of each block is ignored. This weakness is improved in Adaptive-HGFDR by making each block adjust the compression parameters adaptively according to the compression error to achieve the balanced distribution of error. Although the Blocked-HGFDR performs substantially better for several slice numbers, the adaptive HGFDR shows less variations. We have

added the corresponding explanations in page 12 line 298~306.

(14). - Figure 6: Are those the numbers of the slices, or the count of slices overall?

We have reproduced the Figure 6. Those are the order of slices.

(15). - Figure 7: What is "height"? Is that the number of layers? variables? What are the differences between different heights? How are you sub-selecting these layers/variables?

The description of Figure 7 has been modified. The layer is corrected as height. For spatial structure of the data is different at different height. There are both continues and abrupted structure changes at different levels for different variables. To make the experiment more comparable, we select the layers randomly. We have added detailed explanation in page 12 line 314~319.

(16). - Figure 7: What type of error is this? - Figure 7: Color may be helpful. line 243: "error limit" should probably be threshold"? Which type of error are you looking at? The calculation of the error has been defined. It is the relative error ratio, the corressponing expression has been modified in page 9 line 225. We have tried the color graph, however, as the level of error is relatively small and density of data points is large, the grey graph can be seen much clear.

(17). - Figure 8: insufficient discussion of compression time – why is this algorithm so much slower than the competition? Especially so much slower that Blocked-HGFDR which barely shows up on the chart and which provides similar compression ratios Due to the continuous adjustment of compression parameters to search for the optimal rank, Adaptive-HGFDR is the most time consuming. Despite this, some optimization strategies, such as the spatiotemporal indexes and the unbalanced block split, can help improving the efficiency of Adaptive-HGFDR. We have added the corresponding expression in page 14 line 347~349ãĂĆ

(18). - Can this method work in situ with the simulation? Why/why not? YesïijŇthe

method can also suitable for in situ observation data, even the data is sparse , also the data fusion and data synthesis. we have added some extended discussion and potential future directions in the conclusion section.

(19). - Under which conditions does the proposed algorithm perform better? Under which conditions are different algorithms better?

This work focuses on the data compression with the uniform distribution of compression error distribution, which has significant advantages over other methods. Since this method requires constant adjustment of compression parameters, other methods are more effective if you focus on compression time. For the data with strong heterogeneity, because the proposed method divides it into local data block with relatively balanced dimension, it can better use tensor decomposition to capture data characteristics to achieve more accurate data approximation, and it also has advantages in compression ratio. However, for the data with weak heterogeneity, traditional methods can also achieve good compression perfermance due to the gradual change of the characteristics of the data itself, and may have more advantages in compression time.

(20). - Can this algorithm be applied to other types of data? Why/why not? Yes, the algorithm has already been applied to sensor data time series. With any data that can be represented as a tensor, it can be compressed with this method. We have also tried to extend the method to be fit for the irregular data that has arbiter boundaries or sparse data. From the perspective of mathematical foundation, tensor can not only support the multidimensional structure but also can detect the multidimensional coupling feature. Besides that, the tensor can support many kinds of unstructured multidimensional data with a strict mathematical theory. The current main problem is how to construct the concise and efficient algorithm, and found, validation, and solve the core science and technology problem of tensor compression in the practical application. we believe that tension-based spatiotemporal data compression must be an important research direction in the future data management of earth system models. We have added the extend expression in discussion section.

---

## Referee Report (RR1)

This paper presents a method that extends previously published work on Blocked-HGFDR to achieve better lossy compression – both in terms of the distribution of residuals as well as compression ratios. This new method is called Adaptive-HGFDR. The paper includes results from compression experiments to justify the claims about the method.

Although the manuscript has improved since the last revision, many of the previously pointed-out issues remain, and I also have some concerns not previously raised.

Previously pointed-out issues that were not convincingly addressed yet, in my opinion:
1. References still seem off: e.g. Anon, 2011; Of and Acm, 2000; Anon, 2013; None, 1970; Text contains (Springer, 2011) but not the References section; Diffenderfer (2019 a) vs Diffenderfer (2019 b) in the text when there is only one Diffenderfer et al. in the References section
2. An exhaustive list of all outstanding language issues would be too big to list out here. I have included a subset of minor language-related corrections in minor issues, but perhaps the authors should use an "autocorrect tool" to list out all the issues (e.g. I use Grammarly for this purpose).
3. Line 54: "For the file-based compression method, it is difficult to arbitrarily adjust the compression parameter according to the given compression error." This is related to something previously pointed out as not true - I still think this is not true - see e.g. https://github.com/LLNL/H5Z-ZFP
4. Line 73: "...ZFP (Diffenderfer et al., 2019b) are typical methods that use the[sic] feature prediction to achieve lossy compression." Also pointed out in previous reviews that it is not clear why this is being called feature prediction. I looked up the cited paper and it doesn't mention the word "feature". So where is this insight from?
5. The captions for all figures should be expanded so that the reader can answer the questions of "What is going on in this figure?", "What does that mean?", "Why should I care?" are answered right there in the caption, i.e. without having to read the full text.
6. Repeating a previous reviewer's comment: "Not all ESMD data is high-precision. In fact, it is typical that calculations are done in double precision, but that data is output in single precision (e.g., for CESM). What is the precision of the data that you are compressing?" While this was answered in the rebuttal document, but the manuscript was not updated correspondingly (I did a quick Ctrl+F for the word "precision")
7. The above is one example where the authors responded to a comment in the rebuttal but the manuscript was not updated correspondingly, but there are many more. I would advise the authors to go through all the comments from the previous round and make sure the comments are addressed in the manuscript text and not just the rebuttal document.

Issues not previously pointed out but would be nice to fix regardless:
1. The motivation for why a uniform distribution of compression errors is something to strive for is not convincing for me. I understand that lines 36-39 are attempting to do this but I honestly can't follow the line of reasoning. Since this is the core "Why should I care?" of this paper, I think the authors would do well to explain this better.
2. The paper would be easier to read if it had a clear "Contributions" section. As I understand it, the delta of the method described here, as compared to Blocked-HGFDR is that a) each block can have its own rank b) a proposed method for

calculating the optimal rank per block. I think the readers would appreciate having this spelled out towards the beginning of the paper.

3. The "fast search algorithm" described in Definition 4 (line 194) appears to be stated as an original contribution in this paper. To me, this appears to be a rephrasing of "Binary Search", a method that is commonly used in this space. In fact, Grasedyk et al, cited here, also uses this algorithm. The authors would do the reader a favour by making this clearer – either by clearly stating that this is a description of Binary Search, or by clarifying the difference between the stated method and Binary Search.

4. Assuming that I'm not mistaken in the above comment, I'm not convinced that Rouillier et. al. is the best article to cite for the log(n) complexity of Binary Search.

5. Line 55: "For the error truncation-based[sic] compression, the distribution of floating-point precision of ESMD is not uniform, which could lead to the unevenly[sic] distribution of compression errors." Either I misunderstood the statement or this is not true. That rounding/truncation errors are approximately uniformly distributed is a well-known result and used in fields from signal processing to machine learning (e.g. https://arxiv.org/pdf/1802.01436.pdf - this paper refers to it as quantization error)

6. Line 60: "To summarize, it is hard to achieve flexible control of the compression ratio and errors for the description-based lossy compression methods." Perhaps as a result of the other concerns I raise elsewhere, but there doesn't seem to be enough justification provided to make this claim.

7. Line 83: "None of these methods considers ESMD as the[sic] high dimensional data with the heterogeneous correlation between different dimensions." I'm not sure about the other methods but ZFP and SZ surely consider higher-dimensional data. In fact, Tao et. al (2017), cited here, talks specifically about multidimensional prediction. Can the authors please clarify why this does not meet their definition of considering ESMD as high dimensional data?

8. Equation 3 is from Yuan et al (2015) and should be cited as such

9. As I understand it, the hardware used here (HP Compaq Elite 8380 219 MT with Intel Core i7-3770 3.4 GHz processors and 8 GB of RAM) is really small compared to what would be used in realistic runs of this problem. I think this should be pointed out since the compression times are an essential result being reported.

10. It's not clear how the data for Figure 3 was obtained. e.g. text says "… zfp algorithm are affected by the tolerance parameter, which is set to 0.5 ". Does that mean all the data points in Fig 3 for zfp were obtained by the same setting? Then what was varied to get a variation of compression ratios?

11. When I attempted to answer the above question myself by looking at the provided code, I realised that the code does not include ZFP anywhere. Please correct me if I'm mistaken in this conclusion. I think the authors should provide the code to reproduce all of the plots in the paper.

Minor issues:
Line 40: "The main idea of ESMD lossy compressions is to eliminate unnecessary information in data to reduce the data size." This is true of any compression, not just ESMD lossy compression. Also compressions->compression.

develope -> develop

exiting -> existing

unbalance  -> imbalance
prominent components  -> principal components?

---

## Author Response (AR2)

Dear Editor and Reviewers:

This is a major reversion of manuscript gmd-2020-124. Thank you for your interest and helpful comments on our paper.

In the revised version, we reorganized our contents, added several important technological details, and extended the experiments and evaluations. The most significant differences of the reversion and original version are listed as follow:

1.**Title:** The title of the paper was changed from "Adaptive lossy compression of climate model data based on hierarchical tensor with Adaptive-HGFDR (V1.0)" to "Lossy compression of earth system model data based on hierarchical tensor with Adaptive-HGFDR (V1.0)".

**2.Research Motivation:** We rewrote the introduction and basic idea part. In the revise manuscript, we removed the discussion on the uniform distribution of compression error, and focus the topic on adequately exploring the spatio-temporal coupling correlations to reduce the compression error. To make this motivation more clear, we reclassify the existing lossy methods as the predictive and transform methods from the perspective of how the data is approximated. We reviewed the hierarchical-tensor based methods have advantages in utilizing the spatio-temporal coupling correlations to approximate the original data, because they treat all dimensions as a whole, largely reducing the information loss in compression.

Additionally, assign each data block is assigned the independent compression parameter to better capture the local variation of the coupling correlations to improve the approximation accuracy.

**3.Basic Idea:** We developed our method based on the comprehensive consideration of ESMD characteristics. ESMD

have multiple variables with multidimensional structures and the coupling relation, the data distributions along different dimensions of the ESMD are always unbalanced, and the acceptable error of different variables in ESMD is different. Thus, we develop our method form the following perspectives. Firstly, an ideal lossy compression should have the simple parameter and the parameter should be selected adaptively for the acceptable error range of different variables. Secondly, the original data should be divided into a series of local data with more balanced size to reduce the effect of the dimensional unbalance of ESMD. Additionally, the local data in ESMD should have the independent compression parameter to capture the local variation of the multidimensional coupling correlation to improve the approximation accuracy. With these ideas, we developed Lossy compression of earth system model data based on hierarchical tensor with Adaptive-HGFDR.

**5.Experiments:** In the experiments section, we added additional experiments with the method SZ, considering that SZ

may cause the data inconsistency of compression methods, when the data are extracted and analyzed through different orders of dimension combinations. Thus, to verify that the proposed compression method is unrelated to the data organization order, different variables are selected and organized with different orders. Then the advanced predict method SZ and the proposed method are applied to these reorganized data to realize the lossy compression, and the dimensional distributions of compression errors are used to explore the relevance of the method to the data organization order.

To improve the language expressions, we have carefully checked and modified the manuscript accordingly, we also provide a detailed response as follow. We hope this time our paper will meet the high standard criteria of the Geoscientific

Model Development.

We have highlighted the changes in the revised manuscript in MS Word, and detailed responses to the comments are listed as follow:

Referee 2

This paper presents a method that extends previously published work on Blocked-HGFDR to achieve better lossy compression–both in terms of the distribution of residuals as well as compression ratios. This new method is called

Adaptive-HGFDR. The paper includes results from compression experiments to justify the claims about the method.

Although the manuscript has improved since the last revision, many of the previously pointed-out issues remain, and I also have some concerns not previously raised. Previously pointed-out issues that were not convincingly addressed yet, in my opinion:

**(1). References still seem off: e.g. Anon, 2011; Of and Acm, 2000; Anon, 2013; None, 1970; Text contains (Springer,**

**2011) but not the References section; Diffenderfer (2019 a) vs Diffenderfer (2019 b) in the text when there is only one**

**Diffenderfer et al. in the References section.**

We have carefully corrected all the references.

**(2). An exhaustive list of all outstanding language issues would be too big to list out here. I have included a subset of**

**minor language-related corrections in minor issues, but perhaps the authors should use an "autocorrect tool" to list**

**out all the issues (e.g. I use Grammarly for this purpose).**

To improve the language expressions, we have carefully checked and modified the manuscript accordingly.

**(3). Line 54: "For the file-based compression method, it is difficult to arbitrarily adjust the compression parameter**

**according to the given compression error." This is related to something previously pointed out as not true - I still**

**think this is not true - see e.g. https://github.com/LLNL/H5Z-ZFP**

The review about the existing methods in introduction part has been rewrote. These lossy methods have been classified as the predictive and transform methods from the perspective of how the data is approximated. The improper statement about the file-based compression method has been deleted.

**(4)Line 73: "...ZFP (Diffenderfer et al., 2019b) are typical methods that use the[sic] feature prediction to achieve lossy compression." Also pointed out in previous reviews that it is not clear why this is being called feature prediction. I looked up the cited paper and it doesn't mention the word "feature". So where is this insight from?**

The review about the existing methods in introduction part has been rewrote. These lossy methods have been classified as the predictive and transform methods from the perspective of how the data is approximated. The ZFP is classified as the one of the advanced predictive methods.

**(5)The captions for all figures should be expanded so that the reader can answer the questions of "What is going on in this figure?", "What does that mean?", "Why should I care?" are answered right there in the caption, i.e. without having to read the full text.**

All figures have been reproduced, and the detailed captions for all figures have be expanded.

**(6)Repeating a previous reviewer's comment: "Not all ESMD data is high-precision. In fact, it is typical that calculations are done in double precision, but that data is output in single precision (e.g., for CESM). What is the precision of the data that you are compressing?" While this was answered in the rebuttal document, but the manuscript was not updated correspondingly (I did a quick Ctrl+F for the word "precision")**

Yes. The original data we used is double precision. We first process the data into single precision, and then compress it with the proposed method. We have added the corresponding explanation in page 8 line 219.

**(7)The above is one example where the authors responded to a comment in the rebuttal but the manuscript was not updated correspondingly, but there are many more. I would advise the authors to go through all the comments from the previous round and make sure the comments are addressed in the manuscript text and not just the rebuttal document. Issues not previously pointed out but would be nice to fix regardless: The motivation for why a uniform distribution of compression errors is something to strive for is not convincing for me. I understand that lines 36-39 are attempting to do this but I honestly can't follow the line of reasoning. Since this is the core "Why should I care?" of this paper, I think the authors would do well to explain this better.**

In this revised version. We totally rewrote the motivation. The spatio-temporal coupling correlations exist in ESMD, which increases the difficulties in accurately approximating data in lossy compression, thus reduces the compression performance.

Therefore, we removed the discussion on the uniform distribution of compression error, and we focuses on adequately exploring the spatio-temporal coupling correlations to reduce the compression error. Since the multidimensionality and heterogeneity are the natural attributes of ESMD, we further focus on constructing the lossy compression method that integrates both global and local spatio-temporal coupling correlations from the perspective of multiple dimensions. With this idea, we developed Lossy compression of earth system model data based on hierarchical tensor with Adaptive-HGFDR.

**(8)The paper would be easier to read if it had a clear "Contributions" section. As I understand it, the delta of the**

**method described here, as compared to Blocked HGFDR is that a) each block can have its own rank b) a proposed**

**method for calculating the optimal rank per block. I think the readers would appreciate having this spelled out**

**towards the beginning of the paper.**

We have strengthened the contribution in the abstract, introduction and conclusion parts.

We developed an adaptive lossy compression method based on Blocked-HGFDR and improve Blocked-HGFDR from the following perspectives. Firstly, the original data are divided into a series of data blocks with more balanced size to reduce the effect of the dimensional unbalance of ESMD. Then based on the mathematical relationship between the compression parameter and compression error in Blocked-HGFDR, the control mechanism is developed to determine the optimal compression parameter for the given compression error. By assigning each data block independent compression parameter,

Adaptive-HGFDR can capture the local variation of multidimensional coupling correlations to improve the approximation accuracy.

**(9). The "fast search algorithm" described in Definition 4 (line 194) appears to be stated as an original contribution in**

**this paper. To me, this appears to be a rephrasing of "Binary Search", a method that is commonly used in this space.**

**In fact, Grasedyk et al, cited here, also uses this algorithm. The authors would do the reader a favour by making this**

**clearer – either by clearly stating that this is a description of Binary Search, or by clarifying the difference between**

**the stated method and Binary Search.**

With the constructed controlling mechanism, the binary search algorithm is adopted to find the optimal parameter for the data block. We have corrected the corresponding expression in page 7 line 203~206.

**(10)Assuming that I'm not mistaken in the above comment, I'm not convinced that Rouillier et. al. is the best article**
**to cite for the log(n) complexity of Binary Search.**

The reference article has been replaced.

**(11) Line 55: "For the error truncation-based[sic] compression, the distribution of floatingpoint precision of ESMD**
**is not uniform, which could lead to the unevenly[sic] distribution of compression errors." Either I misunderstood the**
**statement or this is not true. That rounding/truncation errors are approximately uniformly distributed is a well-**
**known result and used in fields from signal processing to machine learning (e.g. https://arxiv.org/pdf/1802.01436.pdf -**
**this paper refers to it as quantization error). Line 60: "To summarize, it is hard to achieve flexible control of the**
**compression ratio and errors for the description-based lossy compression methods." Perhaps as a result of the other**
**concerns I raise elsewhere, but there doesn't seem to be enough justification provided to make this claim.**

The review of the existing methods has been rewrote. The inappropriate expression has been revised.

**(12) Line 83: "None of these methods considers ESMD as the[sic] high dimensional data with the heterogeneous**
**correlation between different dimensions." I'm not sure about the other methods but ZFP and SZ surely consider**
**higher-dimensional data. In fact, Tao et. al (2017), cited here, talks specifically about multidimensional prediction.**
**Can the authors please clarify why this does not meet their definition of considering ESMD as high dimensional data?**

Generally, ESMD is the spatio-temporal data with coupling correlations among multiple dimensions. However, most of the
current existing lossy compression methods, including both predictive and transform lossy compression methods, integrate
the spatio-temporal coupling correlations to the data approximation on the foundation of mapping multidimensional data into
low dimensional vector or matrics. Few of these methods directly process multidimensional ESMD as a whole, which may
destroy the multidimensional coupling correlations that largely affect the approximation accuracy in lossy compression. We
have corrected the corresponding expression in the introduction and conclusion sections.

**(13). Equation 3 is from Yuan et al (2015) and should be cited as such**
We have added the corresponding reference in page 7 line 189.

**(14). As I understand it, the hardware used here (HP Compaq Elite 8380 219 MT with Intel Core i7-3770 3.4 GHz**
**processors and 8 GB of RAM) is really small compared to what would be used in realistic runs of this problem. I**
**think this should be pointed out since the compression times are an essential result being reported.**

The proposed compression method is for the model analysis for the end-user, more than the model developer, this is why we choose to conduct the experiment on PC. The experimental results also show that the proposed method can support the lossy compression of ESMD on the ordinary PCs both in terms of the space occupation and compression time. We have added the corresponding expression in introduction and conclusion section.

**(15). It's not clear how the data for Figure 3 was obtained. e.g. text says "… zfp algorithm are affected by the**

**tolerance parameter, which is set to 0.5 ". Does that mean all the data points in Fig 3 for zfp were obtained by the**

**same setting? Then what was varied to get a variation of compression ratios?**

In ZFP, the key parameter is the tolerance. For the given compression error, we conduct the simulation experiments with many random tolerances, and then the ideal tolerances is achieved when the corresponding compression errors are close to the given compression errors. Thus, the tolerance parameters are 0.05, 0.3, 0.5, 3.8 and 10. The detail statement about the parameter of ZFP are added in page 12 line 300~302.

**(16). When I attempted to answer the above question myself by looking at the provided code, I realised that the code**

**does not include ZFP anywhere. Please correct me if I'm mistaken in this conclusion. I think the authors should**

**provide the code to reproduce all of the plots in the paper.**

The code of the algorithm in this work is provided in the form of hyperlink in page 19 line 447~448.

**(17)Minor issues:**

**Line 40: "The main idea of ESMD lossy compressions is to eliminate unnecessary information in data to reduce the**

**data size." This is true of any compression, not just ESMD**

**lossy compression. Also compressions->compression.**

**develope -> develop**

**exiting -> existingunbalance -> imbalance**

**prominent components -> principal components?**

we have corrected the corresponding expression.

Referee 2

**The paper is in good shape, it just needs some light editing:**

**Line 205: typo "coressponding"**

**Line 214-215: is this supposed to be an in-line reference?**

**Line 235: typo "whcih"**

**Line 245: "and" in odd position in "4, 16, 64, and 128, 256"**

**Line 253: typo "Bedsides"**

**Line 310: space in wrong position in "slices( the"**

**Line 332: typo "Form"**

**Line 339: "figure" should be capitalized**

we have corrected the corresponding expression.

[revised manuscript text omitted]

(a) Original data    (b) Compressed data with 4 blocks    (c) Compressed data with 16 blocks (d) Compressed data with 64 blocks    (e) Compressed data with 128 blocks    (f) Compressed data with 256 blocks

**Figure 3. Original data and compressed data with different block counts. (a) The original data; (b) the compressed data when data**
**count is 4; (c) the compressed data when data count is 16; (d) the compressed data when data count is 64; (e) the compressed data**
**when data count is 128;(f) the compressed data when data count is 256.**

**4.3 Comparison with traditional methods**

**4.3.1 Comparison with SZ**

In order to verify that the proposed compression method is unrelated with the data organization order, we select three variables $\{\text{SOLIN, TREFMXAV,FSNTC}\} \in \mathbb{R}^{1024 \times 512 \times 221}$ in ESMD. For each variable, we organize the data with different orders as $\{221 \times 512 \times 1024,\ 512 \times 1024 \times 221,\ 1024 \times 512 \times 221\}$. Then, the SZ and the proposed method are applied to the data to realize the lossy compression. The error distributions of different compression results in the corresponding dimension are shown in the Figure 4.

[Figure]

**Figure 4. The compression error distribution along different dimensions. (a) The compression error distribution along latitude for**
**SOLIN. (b) The compression error distribution along latitude for TREFMXAV. (c) The compression error distribution along**
**latitude for FSNTC.**

[revised manuscript text omitted]